# Taste Masking of Dexketoprofen Trometamol Orally Disintegrating Granules by High-Shear Coating with Glyceryl Distearate

**DOI:** 10.3390/pharmaceutics16020165

**Published:** 2024-01-24

**Authors:** Ilaria Chiarugi, Diletta Biagi, Paolo Nencioni, Francesca Maestrelli, Maurizio Valleri, Paola Angela Mura

**Affiliations:** 1Department of Chemistry “Ugo Schiff” (DICUS), University of Florence, Via Ugo Schiff 6, 50019 Florence, Italy; francesca.maestrelli@unifi.it (F.M.); paola.mura@unifi.it (P.A.M.); 2Menarini Manufacturing, Logistic and Services s.r.l., Via Rosolino Pilo 4, 50131 Florence, Italypnencioni@menarini.it (P.N.); mvalleri@menarini.it (M.V.)

**Keywords:** high-shear coating, taste masking, orally disintegrating granules, glyceryl distearate, hot-melt coating, design of experiments, dexketoprofen trometamol

## Abstract

Orally disintegrating granules (ODGs) are a pharmaceutical form commonly used for the administration of NSAIDs because of their easy assumption and fast dispersion. The development of ODGs is not easy for drugs like dexketoprofen trometamol (DXKT), which have a bitter and burning taste. In this work, high-shear coating (HSC) was used as an innovative technique for DKXT taste masking. This study focused on coating DXKT granules using the HSC technique with a low-melting lipid excipient, glyceryl distearate (GDS). The HSC technique allowed for the coating to be developed through the thermal rise resulting from the friction generated by the granules movement inside the equipment, causing the coating excipient to soften. The design of the experiment was used to find the best experimental coating conditions in order to gain effective taste masking by suitably reducing the amount of drug released in the oral cavity. The influence of the granule dimensions was also investigated. Coating effectiveness was evaluated using a simulated saliva dissolution test. It was found that low impeller speed (300 rpm) and a 20% coating excipient were effective in suitably reducing the drug dissolution rate and then in taste masking. The coated granules were characterized for their morphology and solid-state properties by SEM, BET, XRPD, DSC, and NIR analyses. A human taste panel test confirmed the masking of DXKT taste in the selected batch granules.

## 1. Introduction

Patients’ compliance with oral drug assumptions is influenced by many aspects, including pharmaceutical form, daily dose, size, and ease of swallowing, with the last aspect being of particular significance for pediatric and elderly patients [1].

Orally disintegrating granules (ODGs) are multiparticulate dosage forms where the drug dose is fractionated into multiple small-sized granules and packaged into single-dose sachets named stick packs. These are becoming more and more used for their ease of assumption and fast dispersion within the oral cavity without requiring water ingestion [2]. These aspects are particularly important in the case of pain medications, allowing their prompt intake “on the go”, anywhere and at any time, and providing a fast onset of action [3]. However, developing such formulations for drugs with a bitter and burning taste, like dexketoprofen trometamol (DXKT), can be a real challenge due to issues of poor acceptability from patients [4,5].

The palatability of a drug is defined by the European Medicines Agency (EMA) as “the overall appreciation of an (often oral) medicinal product in relation to its smell, taste, aftertaste and texture (i.e., feeling in the mouth), determined by the characteristics of the active substance, the way the active substance is formulated into a finished medicinal product, and by the characteristics of the excipients”. Regulatory authorities, such as the EMA, FDA (Food and Drug Administration), and ICH (International Council of Harmonization), emphasize the importance of palatability and taste masking in drugs for children and older adults [6,7]. Poor palatability can negatively affect patient compliance and, consequently, the drug’s efficacy, especially for bitter-tasting drugs like NSAIDs and antibiotics. To improve palatability, pharmaceutical companies and researchers use various taste-masking techniques, such as mixing with sweeteners or bitterness inhibitors, or preventing the release of the unpleasant drug taste in the oral cavity by complexation, solid dispersions, or film coatings [8,9].

The latter technique is considered an efficient method for masking bitter tastes by hindering the release of the unpleasant drug taste in the mouth, so far as it allows to meet the drug release requirement for the given dosage form, i.e., for example, the rapid release in the stomach [10]. Coatings can be obtained with different techniques, like fluid bed coating (FBC) and high-shear coating (HSC).

FBC is the most used technique for this purpose, and the applied coating may be a polymer (film coating) or a lipid (hot melt coating). In the first case, the coating agent is dissolved or suspended in a solvent and then sprayed into the powder bed, forming a film on the particle’s surface; then, the solvent evaporates, and it is removed with the fluidizing air stream [11,12]. However, considering the high water solubility of DXKT, the polymer coating would be particularly challenging in the water medium; on the other hand, it is preferable, when possible, to avoid the use of organic solvents for safety and quality reasons. In the second case, the coating agent is applied in its molten state, and it is sprayed into the powder bed without the addition of any solvent and directly solidifies on the surface of the particles. This process is quick and does not require a drying step [13,14]. However, this method is not easily used in the pharmaceutical industry, mostly due to the complexity of the equipment for the heating and spray setup.

HSC is a new technique that allows coatings with a low-melting lipid excipient to be obtained without the use of solvents or external heating sources. In fact, it causes the coating excipient to soften because of the friction generated by the movement of the granules inside the equipment, resulting in a thermal rise, thus allowing the coating to be achieved. Overall, it is a promising method for achieving high-quality coatings with improved properties, as shown by Rosiaux et al., 2018 [15], and it is suitable for different industrial applications.

This research work focused on the use of the HSC technique to obtain granules of DXKT coated with glyceryl distearate (GDS), with the final aim of developing a taste-masked and easy-to-take pharmaceutical dosage form. The proposed technique allows the coating of the drug powder without employing a complex spray setup or an external heating source. This approach aims to simplify the lipid coating process while maintaining its effectiveness. The design of experiment (DoE) was applied to identify and optimize the most significant process parameters of the HSC technique. The effectiveness of the coating was evaluated using the dissolution test in simulated saliva [16,17].

All the coated granules were characterized for their surface area, particle size, and morphological characteristics. A taste panel test was finally performed on the best-coated samples obtained in comparison with the uncoated granular DXKT to assess the actual successful masking of the unpleasant taste of DXKT.

## 2. Materials and Methods

### 2.1. Materials

Dexketoprofen trometamol (DXKT) (Lusochimica SpA, Lomagna, LC, Italy), glyceryl distearate (GDS) (Gattefossè, Saint-Priest, France), colloidal silica (Evonik, Wesseling, Germany), mannitol (Roquette Freres, Beinheim, France), and sucralose (Merk, KGaA, Darmstadt, Germany) were a generous gift from Menarini Manufacturing Logistic and Services s.r.l., Florence, Italy. All other reagents were of technical grade.

### 2.2. Wet Granulation in a High-Shear Mixer (HSM)

Drug powder was subjected to wet granulation with a HSM using the GMXB-LAB mini GRANUMEIST^®^ Bottom Drive (Freund Vector, Marion, IA, USA), equipped with a 6-L mixing bowl. Water was used as a granulating solution. The drug powder was previously mixed with colloidal silica, which was added to limit adhesion between particles during the process [18,19]. To obtain the necessary amount of DKXT granules, various batches of 1 Kg were produced and then collected together. For each batch, 97.5% of the active ingredient and 2.5% of the colloidal silica were processed in the HSM in three stages: powder mixing, wetting, and massing. In the first step, the powders were mixed for 1 min with an impeller speed of 400 rpm and a chopper speed of 600 rpm. During the wetting stage (5 min), water was added at a constant flow rate of 15 mL/min; the amount of water was 75 mL; the impeller speed was set at 400 rpm and the chopper speed at 600 rpm for 3.5 min; then, the chopper speed was increased to 1500 rpm, remaining constant for the rest of the process. The final stage of massing allowed granule growth thanks to the conjunctive action of the impeller (400 rpm) and chopper (1500 rpm). The massing time was 1 min to avoid overwetting phenomena. The wet granules were obtained using the Erweka AR 400 oscillating granulator, equipped with a 1.5 mm net size (GPI Equipment Group, Jefferson Valley, NY, USA). The obtained granules were dried in a static oven (FD 250 F 9 Vismara s.r.l., Vallemare, PE, Italy) at 50 °C for 2 h, until 1.5% of the residual moisture was reached. The moisture was checked using the loss on drying (LOD) test using the moisture analyzer Mettler HR 83 (Mettler Toledo AG, Nänikon, Switzerland). The granules were then screened with a vibrating test sieve (Retsch GmbH, Haan, Germany) to obtain three different granulometric fractions: the biggest one (B), greater than 850 μm; the medium one (M), between 180 and 850 μm; and the smallest one (S), lower than 180 μm. All the batches were stored in a HDPE (high-density polyethylene) drum with a double bag liner and desiccant bags.

### 2.3. Design of Experiments (DoEs)

A design of experiments (DoEs) strategy was performed to find the optimal conditions for an effective coating of the DXKT granules. The JMP 17.2.0 version (SAS Analytics Institute Inc., Cary, NC, USA) tool was used for the generation of the statistical experimental design and evaluation of the results. The significance and validity of the model were evaluated using the ANOVA (analysis of variance) statistical test with a Tukey–Kramer test.

### 2.4. Coating in a High-Shear Mixer

The same HSM apparatus used for granule production was also used for the granule coating. To produce coated granules, DXKT granules and GDS were weighed in different proportions according to the plan generated using the experimental design software JMP 17.2.0 to obtain a mass of 1 kg for each batch. The two-step coating process included a heating phase and a coating phase. During the heating phase, the chopper was off to avoid unwanted granules crushing, and the impeller speed was fixed according to the corresponding DoE run until the target temperature of 42 °C (softening temperature of GDS) was reached. During the coating phase (15 min), the chopper was set at 300 rpm (minimum speed allowed by the instrument), and the impeller speed was reduced to 150 rpm regardless of the value of the heating phase. After obtaining the coated granules, three particle-size fractions were separated as described above (Section 2.2) and used for subsequent studies.

### 2.5. UV-Vis Spectrophotometry

The UV-visible spectrophotometer Uv-1900 (Shimadzu, Kyoto, Japan) was used for the determination of DXKT concentrations in the drug dissolution test in simulated saliva (SS). This analysis was conducted in 1-cm quartz cells. The wavelength of the absorption maximum of DXKT was λ = 258 ± 2 nm. For the calibration line, 10.40 mg of DXKT powder was weighed and made up to volume (100 mL) with deionized water. A total of 10 mL of the bulk solution was taken, and a 1:5 dilution (stock solution) was performed. Solutions of known concentration were obtained from the stock solution by performing five different dilutions (2:10, 3:10, 4:10, 5:10, and 6:10). UV readings of the solutions at a known concentration at λ = 258 nm provided the drug calibration line for the determination of unknown concentrations. It was previously verified that the presence of GDS did not interfere with the drug assay.

### 2.6. Dissolution in Simulated Saliva (SS)

Simulated saliva (SS) buffer at pH 6.8 was prepared by dissolving 2.38 g/L of Na_2_HPO_4_, 0.19 g/L of KH_2_PO_4_, and 8 g/L of NaCl in deionized water, and the desired pH was obtained by adding H_3_PO_4_. The dissolution test in SS was performed by placing a beaker with 50 mL of simulated saliva in a thermostatic bath at 37 °C and adding granules equivalent to 25 mg of drug (a single DXKT dose). A stirrer consisting of a steel bar ending in three fins placed at regular distances from each other was placed in the center of the beaker; the stirrer was connected to a motor and rotated at 200 rpm. At set times of 1, 3, and 5 min, 3 mL of solution were taken with a syringe and filtered through a 0.45-μm filter. A total of 1 mL of the obtained sample was diluted 1:50 and analyzed using UV spectrophotometry to determine the amount of drug in solution.

### 2.7. Solid-State Characterization

Samples of pure DXKT powder, pure GDS powder, and coated and uncoated DXKT granules were characterized at the solid state using the following techniques: X-ray powder diffractometry and differential scanning calorimetry analysis were performed in order to examine whether any change in drug solid-state properties occurred during the granulation process.

X-ray powder diffractometry (XRPD): The instrument used was a theta–theta Bruker D8-advance Powder Diffractometer apparatus (Silberstreifen, Germany) using Cu Kα radiation and a graphite monochromator. The samples were analyzed at room temperature in the 3–30° 2θ range, with the following experimental conditions: step = 0.03-time scan = 1 s, i = 40 mA, V = 40 kV, and λ = 1.54018 Ả.

Differential scanning calorimetry (DSC): Samples (5 to 10 mg) were exactly weighed with a Mettler MX5 microbalance (Mettler-Toledo, Greinfensee, Switzerland) and put in sealed Al pans with pierced lid. Analyses were conducted on a Mettler TA 4000 Stare system (Mettler Toledo, Greifensee, Switzerland) using the following parameters: scanning in a temperature range between 30 and 300 °C, a static air atmosphere, and a scanning speed of 10 °C/min.

Scanning electron microscopy (SEM): SEM was used to examine the morphologies of the coated and uncoated DXKT granules. The samples were analyzed under high vacuum and underwent a preliminary treatment of metallization; that is, they were coated with a metallic film (silver) to better appreciate their morphological characteristics. The samples were analyzed using a FIB-SEM GAIA 3 instrument (Tescan, Brno, Czech Republic).

Specific surface area (SSA): SSA was evaluated with Micromeritics ASAP 2010 equipment (Micromeritics Norcross, GA, USA) according to the BET (Brunauer-Emmett-Teller) method, which involves measuring the amount of inert gas absorbed by a monomolecular layer of the solid under investigation. This was carried out by degassing the sample and then adding controlled amounts of nitrogen gas into a burette connected to an analytical station under vacuum. The nitrogen adsorption isotherm was measured by adding nitrogen gas, and the nitrogen desorption isotherm was measured by withdrawing nitrogen gas. The equilibrium pressure in the burette was measured after each sampling.

Particle size analysis using sieves: Particle size analysis using sieves was used to separate the different fractions of uncoated and coated granules. The used instrument was a vibratory sieve shaker equipped with sieves of 850 μm and 180 μm (Retsch GmbH, Haan, Germany).

Laser diffractometry: The particle size and particle size distribution of the samples were measured through laser diffractometry using Mastersizer 3000 equipment (Malvern Panalytical Ltd., Worcestershire, UK). The operating parameters used were: dispersant = Tegiloxan 3; stirrer speed = 2800 rpm; background measurement time = 15.00 s; sample measurement time = 5.00 s; and light obscuration = 16.48%.

Loss on drying (LOD): LOD values of uncoated DXKT granules were measured following a two-hour stay in an oven at 50 °C; granules with a moisture content of less than 1.5% were considered acceptable. LOD analyses were performed by setting the temperature at 80 °C until the loss of weight was constant for 20 s. Approximately 3 g of granules were weighed to perform the LOD measurement. The results were expressed as percent mass lost (% *w*/*w*) relative to the initial weight. A Mettler HR 83 infrared balance (Mettler-Toledo, Greinfensee, Switzerland) was used.

Near infrared spectroscopy (NIR): NIR spectra were acquired between 900 and 1700 nm in the diffuse reflectance mode (200 scan count and 7.8 ms integration time) using a Micro NIR PAT-W Spectrometer (Viavi Solution Inc., Chandler, AZ, USA). A 99% reflectance sample was used as a reference. The spectral acquisition and data elaboration were performed using Unscrambler X 10.3.1 lite Software (Oslo, Norway). This technique was used as an in-line analysis, with the aim of monitoring the coating process and highlighting any differences.

Data elaboration: For data collection, elaboration, and analysis, Excel and JMP Software 17.2.0 (SAS Institute Inc., Cary, NC, USA) were used.

### 2.8. Human Taste Panel Test

In order to carry out an organoleptic evaluation of the coated granule samples, a panel tasting test was conducted on 8 subjects [20]. The coated granules that seemed to have the best effective coating, based on their dissolution test results, were tested. A panel of 8 adult healthy volunteers experienced in sensory analysis was used for this study. The panel provided their written informed consent to take part in this research and expectorated all samples; thus, ethical approval and specific medical inclusion and exclusion criteria were not required. This research was not considered a medical drug trial, so it was carried out in accordance with the IFST Guidelines for Ethical and Professional Practices for the Sensory Analysis of Foods and not specifically under the Code of Ethics of the World Medical Association (Declaration of Helsinki). The protocol for testing involved the following: Single-dose sachets with a total weight of 300 mg were prepared, containing 46.5 mg of DXKT-coated (or 37 mg DXKT-uncoated) granules equivalent to 25 mg of active substance (a single drug dose), 1 mg sucralose, and granular mannitol q.b. to 300 mg. Four samples were tested in the following order: “placebo” granules as a positive reference (containing all the formulation components without the API), two samples of DXKT-coated granules, and, finally, uncoated granules as a negative reference. All the subjects were instructed to keep the samples in their mouths for not more than 60 s, avoiding any possible powder swallowing, and then spit them, thoroughly rinsing the mouth at the end of each trial. The placebo was tested first to accustom the palate to the sweet and fresh sensations of mannitol and sucralose. Then, all the subjects tested the selected DXTK-coated granule samples and, finally, the uncoated ones, giving each sample a specific score in terms of the onset time of the "burning effect" related to DXKT. After each test, the subjects repeatedly rinsed their mouths with water, and there was 15 min of rest time between each round. The results were evaluated using the ANOVA statistical test with a Tukey–Kramer test.

## 3. Results and Discussion

### 3.1. Preliminary Evaluation

According to the findings in the literature, it is suggested that the active pharmaceutical ingredient (API) to be coated should have a granulometric size that is not excessively fine [15,21,22,23,24]. However, no specific data regarding DXKT were available in the literature. Therefore, an initial attempt has been made to directly coat the DXKT powder with 10% GDS. However, it was unsuccessful, most likely due to the too-fine particle size of the API (Dv90 = 23.70 μm) and to the consequent low interparticle movement. In fact, this resulted in insufficient friction, which did not allow the proper temperature rise for softening the coating excipient and then enabling it to suitably coat the drug [25]. It was therefore necessary to proceed with the production of DXKT granules with a suitable size for coating.

### 3.2. Production and Characterization of DXKT Granules

In order to obtain the necessary amount of DXKT granules, several batches were produced, as described in Section 2.2, and then collected together. The granules were divided into three granulometric fractions: B (Big) greater than 850 μm, M (Medium), between 180 and 850 μm, and S (Small), lower than 180 μm, that were then carefully characterized. SEM images (Figure 1) revealed that M and B fraction granules exhibited similar rather homogeneous oblong shapes, while S fraction granules appeared more irregularly shaped, also showing a considerable amount of dust particles.

Particle size analysis by sieving indicated that fraction B had a yield of 32%, fraction M had a yield of 60%, and fraction S had a yield of 8%. BET values showed that the SSA increased as the granule size decreased, with the B fraction having a value of 0.5 m^2^/g, the M fraction 0.6 m^2^/g, and the S fraction 0.8 m^2^/g.

XRPD and DSC analyses were performed to investigate possible changes in the drug’s solid-state properties that could have been caused by the granulation process.

For such a reason, the XRPD spectra of basic granules, “as such” and after mortar grinding, were compared with both the theoretical spectrum of DXKT and that of the starting DXKT sample (Figure 2). In particular, the grinding of the granules was undertaken to assess whether it produced any modifications in the drug’s crystalline state. The comparison of the spectra revealed that both ground and not-ground granule samples contained the same crystalline phase of DXKT, indicating its ability to maintain its practically unchanged crystalline structure under mechanical treatments. However, a peak at 6.3° 2θ was observed in the DXKT granule spectrum, which was not present in the theoretical spectrum of the salt and instead perfectly corresponded to that of the starting DXKT powder sample. This peak could indicate the presence in the granulate of the dihydrate salt of DXKT, which has a peak at this same 2θ value.

This evidence was further confirmed by the DSC analysis (Figure 3), where the thermal curve of the DXKT-untreated powder exhibited a sharp endothermic peak at 105.24 °C (ΔH = 87.83 J/g), due to the melting of the pure crystalline anhydrous drug, followed by a broad endothermic band observed in the range between about 170 and 250 °C that was ascribed to decomposition phenomena. On the contrary, the thermal curve of the granules showed the presence of three endothermic events: the first one, which peaked at 56.12 °C (ΔH = 35.16 J/g), was attributed to the loss of water of the hydrated salt, while the second one, which peaked at 105.24 °C (ΔH = 60.89 J/g), coincided with the melting of the anhydrous DXKT, and the last one, which ranged between 170 and 250 °C, was imputed, as above, to decomposition phenomena. These results confirmed the presence of a hydrated form of the salt that was not present in the starting sample [26]. The wet granulation process was responsible for the formation of the salt-hydrated form.

NIR spectra are reported in Figure 4. Figure 4A presents the NIR spectra of DXKT and GDS. The DXKT spectrum exhibits three distinct bands that peaked at 1130, 1170, and 1400 nm, while the GDS spectrum shows peaks at 1030, 1200, and 1450 nm. In Figure 4B, the NIR spectrum of DXKT powder is compared to that of the uncoated granules. The granules exhibit a similar spectrum to that of the powder, except in the range of 1410–1560 nm. This discrepancy may be attributed to the presence in the granules of the dihydrate form of DXKT salt, as water absorption occurs in this wavelength range.

The dissolution test in SS was performed to confirm that granulation did not affect the release of DXKT. The % of DXKT released was high and comparable to that of DXKT powder. For example, 95% of DXKT was released after 1 min from fraction M of the granules, reaching 98% after 3 min and 100% after 5 min.

### 3.3. Production of Coated Granules

GDS was selected as the coating agent mainly because of its suitable low melting temperature (40 to 50 °C), safety in use, and proven ability to efficiently mask the taste of drugs by forming a film coat [15,21].

#### 3.3.1. Preliminary Test

A preliminary coating test was performed using fraction B granules and 8% GDS. This test was successful: the required temperature rise (42 °C) was obtained, indicating that the granule size was suitable for coating. The release of DXKT in simulated saliva showed that 74.7% were released at 1 min, 91.2% at 3 min, and 97.9% at 5 min. The coated granules had a slower dissolution rate compared to the uncoated ones; however, the obtained reduction in the dissolution rate was not enough to mask the unpleasant taste of DXKT. Therefore, a DoE strategy was applied to optimize the parameters of the coating process.

#### 3.3.2. Experiments according to the DoE

With the aim of identifying the best experimental conditions for the coating process (heating phase), a DoE was designed and performed. The potential variables for the heating phase include the chopper speed, impeller speed, % GDS, and granule size. During the heating phase, the chopper is turned off to prevent granule fragmentation. The DXKT-uncoated granule’s fraction M (180–850 µm) was chosen as the starting material for coating as it represented the fraction obtained in the highest quantity. It was decided to investigate the effects of the remaining two variables: impeller speed and GDS %, each at two levels. A full factorial design reduced with a D-optimal criterion to four trials was applied. The experimental domain (high- and low-level values) of each factor was determined based on subject matter knowledge.

The evaluated responses were the time to reach 42 °C (temperature required for excipient softening and distribution) and the % of DXKT released in simulated saliva after 1 min. The experimental plan of five experiments, including one center point at intermediate levels of the variables (450 rpm and 15% GDS), and the obtained responses are shown in Table 1.

An inverse relation was found between impeller speed and time to reach 42 °C, which increased as the impeller speed decreased, regardless of the GDS content. On the contrary, the change in the level of the impeller speed clearly affected the drug release rate. In fact, as can also be observed by the results of the dissolution test in simulated saliva, presented in Figure 5, the highest reduction of % drug released was obtained with the highest level of % GDS (20%), but only when it was associated with the lowest impeller speed, as evident by comparing the data of batches 3 and 4.

A predictive model for each response was obtained. The first model, shown in Figure 6, investigates the variation of the time to reach the temperature of 42 °C. ANOVA analysis indicated that the model was valid and significant (*p* < 0.005). The impeller speed was identified as the only significant factor, with two terms included in the model: the main impeller speed term and a quadratic term. The rate of temperature increase was observed to be constant and proportional to the impeller speed. This predictive model was used to estimate the time required for coating processes for various granules.

The second predictive model (Figure 7) was instead developed based on the results of the % of DXKT dissolved after 1 min in simulated saliva.

The model was statistically significant (*p* < 0.05), even though the level of significance was lower than for the previous model (*p* < 0.005). It can be observed that, unexpectedly, the factor % of GDS had a lower influence on the coating degree, evaluated in terms of drug dissolution rate reduction, compared to the impeller speed factor. Probably, a high impeller speed did not allow for a thickening of the coating, favoring instead the formation of a thinner covering.

#### 3.3.3. Additional Experiments

Considering that only one batch was obtained at the impeller speed of 450 rpm and that this parameter was crucial for the coating process, we decided to perform further experiments at this speed. Specifically, we aimed to explore the impact of increasing the percentage of GDS. Previous results showed the greatest reduction in the drug dissolution rate with the highest GDS percentage tested (20% GDS). Therefore, it was deemed appropriate to explore the potential benefits of further increasing the GDS percentage to 25%. Therefore, two additional batches of coated granules were prepared using the M granule fraction, with the GDS amounts set at 20% and 25%, while maintaining a constant impeller speed of 450 rpm (Table 2). The resulting drug-release profiles in simulated saliva were examined and presented in Figure 8. Unexpectedly, the dissolution curves indicated that, at this impeller rate, the percentage of DXKT dissolved after 1 min from the granules coated with different GDS concentrations was consistently around 35%. However, the effect of the coating became apparent after 3 min, and even more after 5 min. Furthermore, the percentage of DXKT dissolved after 1 min was consistently higher than that observed for the granules coated with 20% GDS and an impeller speed of 300 rpm (see run 4 in Table 1). This confirmed that the combination of 20% GDS and an impeller speed of 300 rpm yielded the most favorable results.

Once the appropriate values for the impeller speed and GDS percentage that yielded the best granule coverage in terms of drug dissolution rate reduction (300 rpm and 20% GDS) were established, we considered it important to explore the potential level of coverage achievable with these same parameters on granules of different sizes. Two additional coated granule batches were then prepared, starting from the granule fractions B and S.

As can be seen in Table 3, the production of coated granules starting from different granulometric fractions of the uncoated granules evidenced the existence of an inverse relationship between granule size and temperature rise rate (°C/min) during the coating process. In particular, the time required for the S fraction to reach the GDS softening temperature and then allow the granules to coat was significantly longer compared to the other fractions. Moreover, the % of DXKT dissolved at 1 min from this granulometric fraction was very high (about 88%) and only a little lower than that from the uncoated granules, indicating a poor coating effect.

The poor effectiveness of the coating obtained starting from the S fraction is also evident from observing Figure 9, where the DXKT dissolution profiles in simulated saliva from coated granules obtained starting from the three considered granule fractions under the selected conditions (impeller speed 300 rpm and 20% GDS) are compared.

It was concluded that the granule fraction S was unsuitable for obtaining an adequate coating since the lower dimensions of the granules were too small to allow for an effective softening of GDS and gave rise to a larger surface area to be covered. This confirms the reason for the failure of the preliminary coating test performed on the active ingredient in its original particle size, which was too fine.

On the contrary, a good degree of coating can be achieved with the M and particularly with the B fractions, as proved by the marked reduction in the drug dissolution rate. It appears that larger granules result in greater friction between particles, leading to more homogeneous and faster coating, while smaller granules result in less friction between particles, giving rise to longer process times and inhomogeneous coating.

The tests conducted demonstrated that a low impeller speed and an adequately sized uncoated granule fraction (>180 μm) are crucial factors in achieving a coating sufficient to slow down the release of the active ingredient in simulated saliva. Therefore, keeping the impeller speed at the lowest value (300 rpm), M and B granulometric fractions were selected to investigate the effect of increasing to 25% of the GDS. The results are shown in Table 4 and Figure 10, in comparison with those obtained from the batches produced with the same granule fractions and 20% GDS.

The findings indicated that, for both granule fractions, increasing from 20 to 25% of the coating excipient only slightly reduced the excipient softening time as well as the % drug dissolved at 1 min. Therefore, it was concluded that there were no advantages to increasing the percentage of coating agent, and the 20% coated granules from the M and B granule fractions (batch 04 and 09) were selected for further characterization.

### 3.4. Characterization of Coated Granules

The NIR spectra obtained from the coated granules produced starting from the uncoated M fraction granules at a fixed impeller speed (300 rpm) and different percentages of coating excipient (10–20 and 25%) are shown in Figure 11. As can be seen, the coating process can be followed by controlling the intensity of the peak at 1205 nm, which increases with increasing the GDS percentage.

Selected SEM images of fraction B uncoated granules and their corresponding coated granules (batch 04) are shown in Figure 12. As can be seen, both kinds of granules presented irregular shapes, but the surface of the coated samples appeared smoother than that of the uncoated ones, which looked wrinkled.

The SSA values of the three uncoated granule fractions (S, M, and B) and of their corresponding coated granules produced using an impeller speed of 300 rpm and 20% GDS are shown in Table 5. As can be seen, the values were rather low and, as expected, gradually increased as the granule size decreased. Based on the isotherms obtained by BET analysis, it was possible to determine the specific volume of N_2_ adsorbed in pores of less than 50 nm to assess the porosity of the materials. In the case of uncoated granules, it was less than 0.1 cm^3^/g, allowing us to define them as nonporous materials. A slight increase was instead observed for the coated granules, which could be attributed to the formation of small aggregates that created a type of artificial porosity. In any case, the values obtained were minimal, leading to the conclusion that the coated granules were also nonporous materials.

Figure 13A shows the XRPD spectra of the coated granules produced from the M and granulometric fractions under identical conditions, i.e., an impeller speed of 300 rpm and 20% GDS. From the observation of the overlapped diffraction patterns of the coated granules, it was found that both samples contained an anhydrous phase and a dihydrate phase of DXKT salt (as already observed for the uncoated granules), as well as an unspecified amount of amorphous material that was likely due to the presence of GDS (Figure 13B). These findings suggest that both types of coated granules exhibited comparable characteristics with regard to the identified DXKT phases and the amorphous material present.

The DSC curves of the coated granules obtained under optimal conditions (an impeller speed of 300 rpm and 20% GDS) starting from the M and B granulometric fractions are shown in Figure 14. Similarly to that previously observed for uncoated granules (see Figure 3), the thermal curves of both coated granules displayed three endothermic effects, where the first one, which peaked at around 55 °C, can be attributed as above to the loss of water, confirming the presence of the hydrate form of the salt, as indicated by XRPD analysis. An overlapping od this band due to the melting of GDS (T_fus_ 56 °C) could also be hypothesized. However, the enthalpy values of this band in the case of coated granules (31.38 J/g for batch 04 and 31.74 J/g for batch 09, respectively) were very similar to those found for uncoated granules (35.16 J/g) not containing GDS. This finding therefore seemed to confirm that GDS was present in the granules in the amorphous state, as indicated by XRPD analysis. On the other hand, the second DSC peak, which was observed at around 106 °C, is due to the melting of the anhydrous DXKT salt, and the final broad band between 170 °C and 250 °C is due to decomposition phenomena. These results indicated that both types of coated granules possess similar thermal properties and contain the same phases of DXKT salt, and they definitely confirmed that the coating process did not cause any change in the drug solid phases.

### 3.5. Taste Panel Test

An effective suppression of the unpleasant taste of a drug is of great importance for achieving good patient compliance, particularly in the case of orally dispersing dosage forms such as the ODGs. To evaluate whether our goal of masking the DXKT bitter taste by HSC coating with GDS has been actually achieved, the selected coated granules obtained from the granulometric fractions M and B under optimal conditions (20% GDS and 300 rpm impeller speed) were evaluated through a human panel test performed on eight adult volunteers using as a positive reference a “placebo” (granules containing all the formulation components without the API) and as a negative reference the uncoated drug granules. The results obtained in the taste panel test, expressed as the arithmetic mean ± the standard error of the onset time (the residence time in the oral cavity before the onset of the burning sensation due to DKXT), are presented in Table 6.

The data were statistically analyzed using ANOVA, performing a statistical comparison with the Tukey–Kramer test (Figure 15). The statistical analysis indicated that there was a significant difference (*p* < 0.0001) between the uncoated DKXT granules and both coated granule samples, proving the effectiveness of the coating in masking the unpleasant taste of the active ingredient in both batches of coated granules. Moreover, there was no significant difference between the two kinds of coated granules. The results were visually represented using circles, with different colors indicating a significant difference in the outcomes, while circles of the same color indicated no significant differences. This graphical representation provided a clear and intuitive way to interpret the results, allowing for a quick identification of statistically significant variations between groups.

## 4. Conclusions

In conclusion, the present study successfully achieved the desired objective of developing coated granules of DXKT with improved taste, enabling the formulation of ODGs for a more convenient drug administration.

The application of the HSC technique, utilizing GDS as the coating excipient, proved to be a valuable and straightforward method for masking the unpleasant organoleptic characteristics of DXKT. The innovative coating technique, relying on the friction-induced thermal increase, demonstrated its effectiveness in softening the low-melting coating agent and achieving successful coating of DXKT granules.

It was found that a granule size of over 180 µm was necessary for achieving optimal coating results. The degree of granule coating was primarily influenced by the impeller rotation speed and the particle size distribution of the basic granules. A coating excipient percentage of at least 20% was essential for an effective coating, while no additional improvements were observed at higher percentages. By employing an intermediate Medium (M) or Big (B) particle size fraction of the basic granules, along with a low mixing speed (300 rpm) and a 20% coating excipient percentage, it was possible to achieve an efficient coverage able to mask the unpleasant taste of DXKT, as confirmed using a final human panel test.

Overall, the findings of this research provide valuable insights into the development of a palatable and easily administrable formulation of DXKT, highlighting the potential for improving patient compliance and satisfaction. The novel coating technique and optimized parameters explored in this study contribute to the advancement of pharmaceutical formulations and pave the way for further research in the field of oral drug delivery systems.

## Figures and Tables

**Figure 1 pharmaceutics-16-00165-f001:**
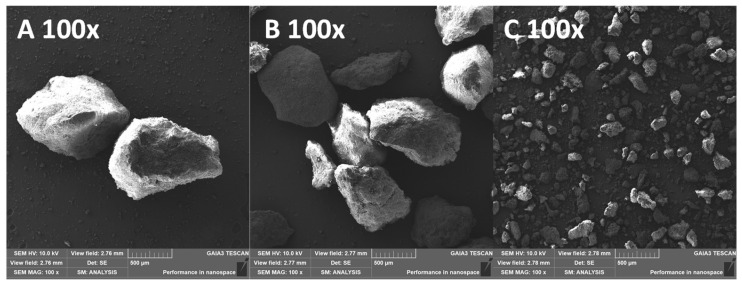
SEM micrographs at 100× magnification of uncoated granules: granulometric fractions Big (**A**), Medium (**B**), and Small (**C**).

**Figure 2 pharmaceutics-16-00165-f002:**
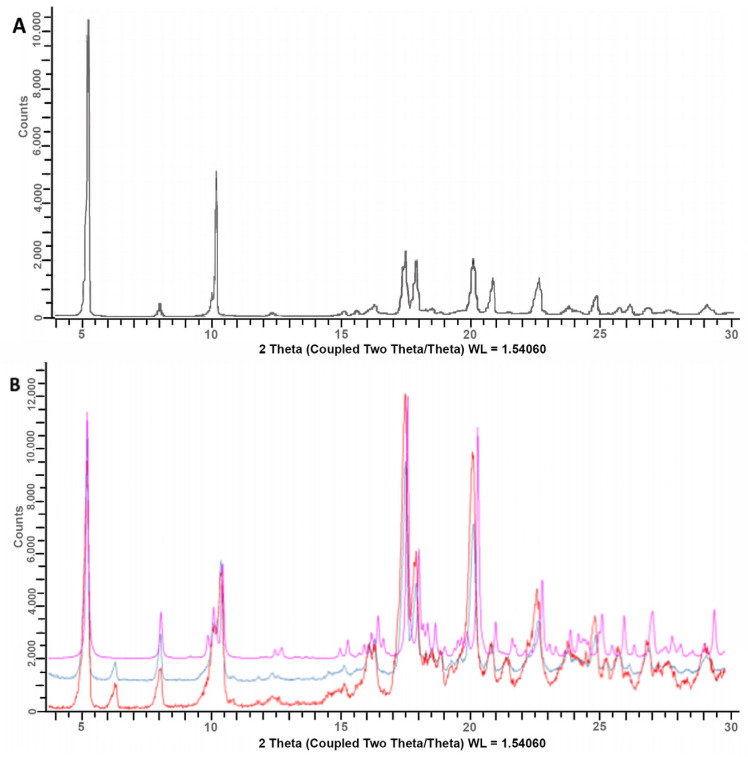
XRPD spectra of the starting DXKT powder sample (**A**) and comparison of spectra of basic granules, “as such” (red) and after mortar grinding (blue), with the theoretical spectrum of anhydrous DXKT (pink) (**B**).

**Figure 3 pharmaceutics-16-00165-f003:**
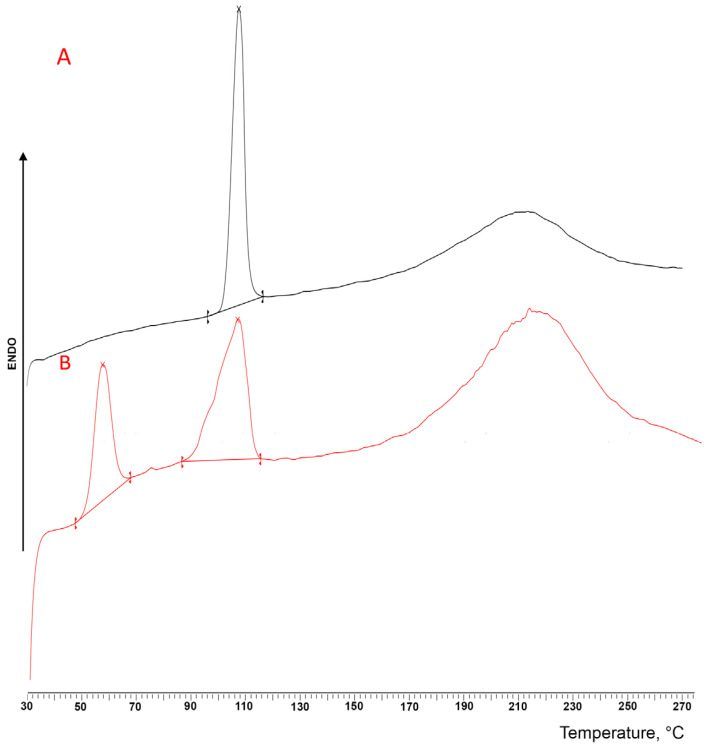
DSC curves of DXKT powder (**A**) and uncoated granules (**B**).

**Figure 4 pharmaceutics-16-00165-f004:**
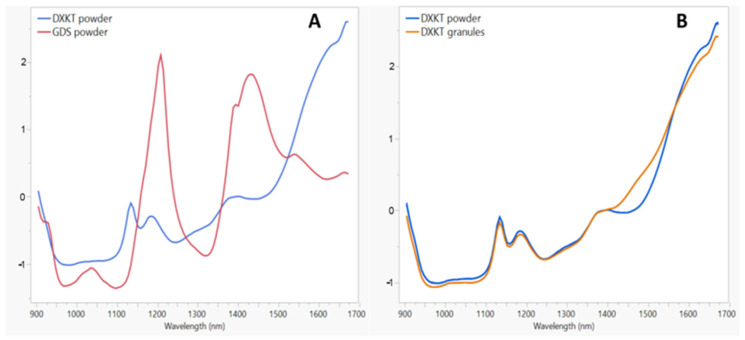
(**A**) NIR spectra of the starting DXKT powder and GDS samples. (**B**) NIR spectra of DXKT powder and DXKT uncoated granules (fraction M).

**Figure 5 pharmaceutics-16-00165-f005:**
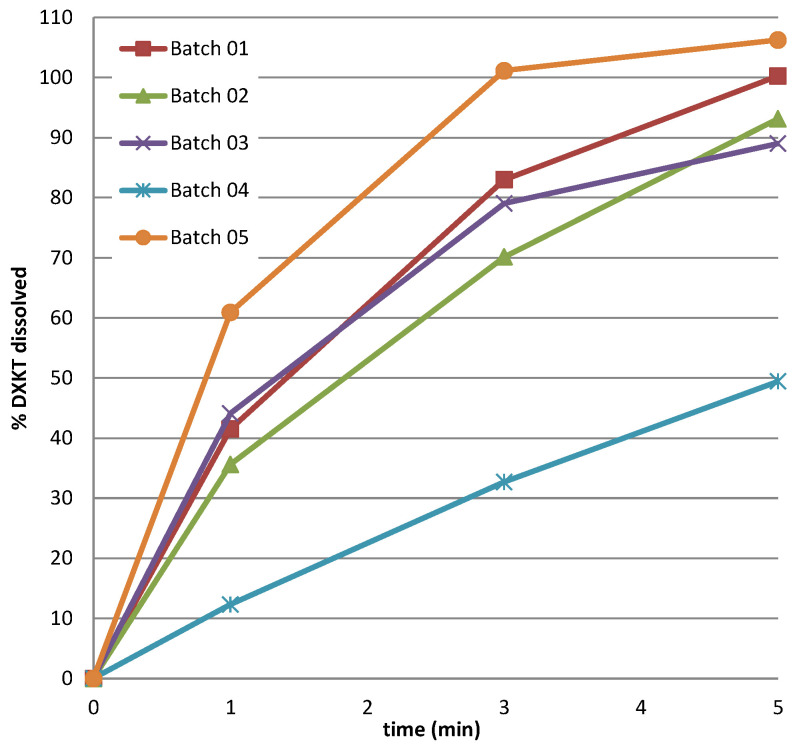
Dissolution rate in simulated saliva of DXKT from coated granules obtained from fraction M granules according to the DoE plan (see Table 1).

**Figure 6 pharmaceutics-16-00165-f006:**
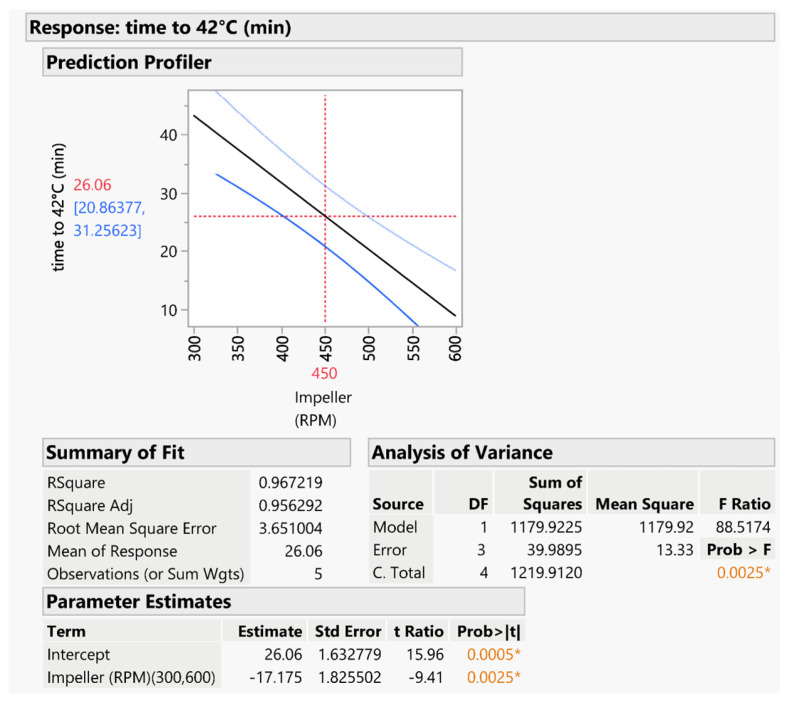
Predictive model for estimating the time to reach a temperature of 42 °C and related statistical evaluation. The asterisk indicates the statistical significance of the model; the orange color is used for *p* values lower than 0.01.

**Figure 7 pharmaceutics-16-00165-f007:**
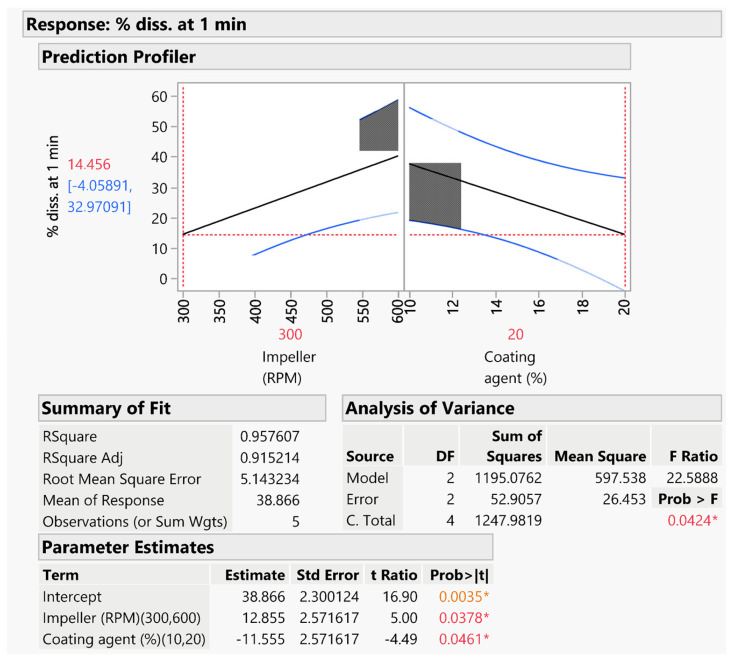
Predictive model for estimating the % of DXKT released from coated granules after 1 min in simulated saliva and related statistical evaluation. The asterisk indicates the statistical significance of the model; the orange color is used for *p* values lower than 0.01, the red color for *p* values from 0.05 to 0.01.

**Figure 8 pharmaceutics-16-00165-f008:**
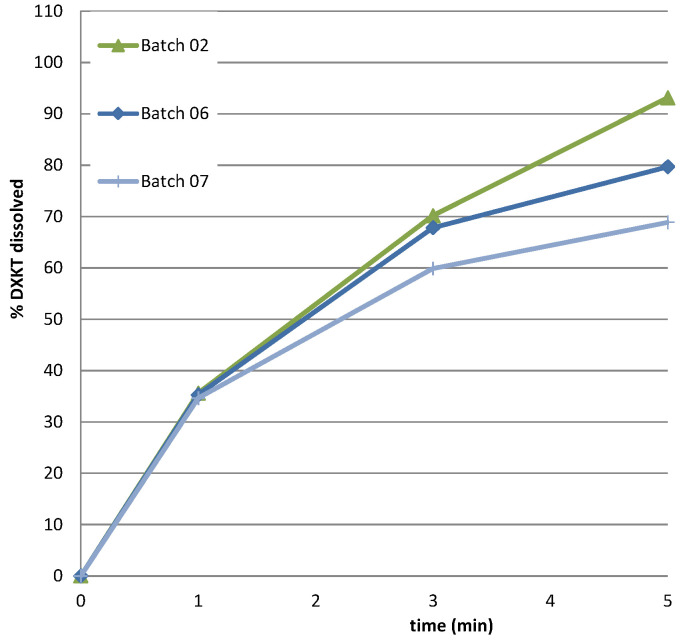
The dissolution rate of DXKT in simulated saliva from coated granules produced starting from fraction M at impeller speed 450 rpm and 15% (batch 02), 20% (batch 06), and 25% (batch 07) GDS.

**Figure 9 pharmaceutics-16-00165-f009:**
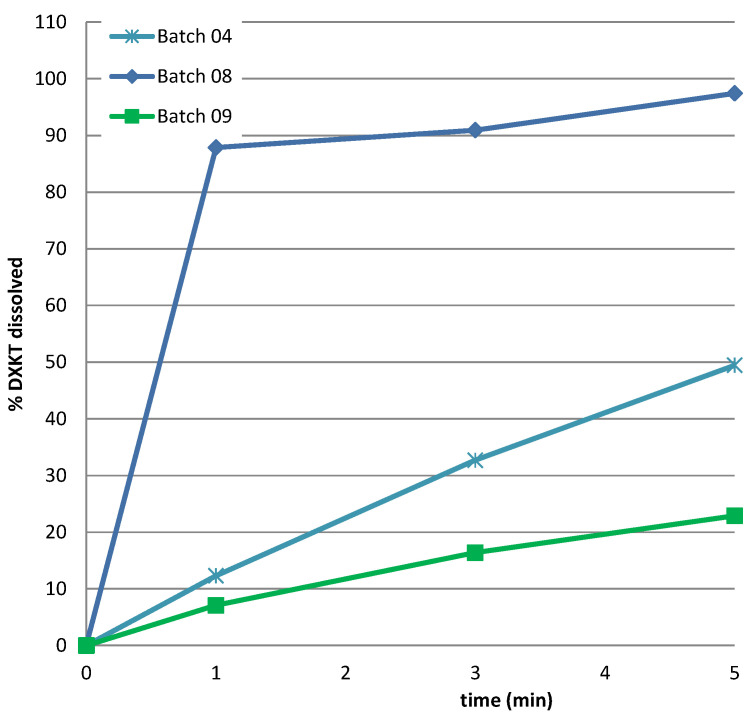
The dissolution rate of DXKT in simulated saliva from coated granules produced starting from fractions S, M, and B under optimal conditions (impeller speed of 300 rpm and 20% GDS).

**Figure 10 pharmaceutics-16-00165-f010:**
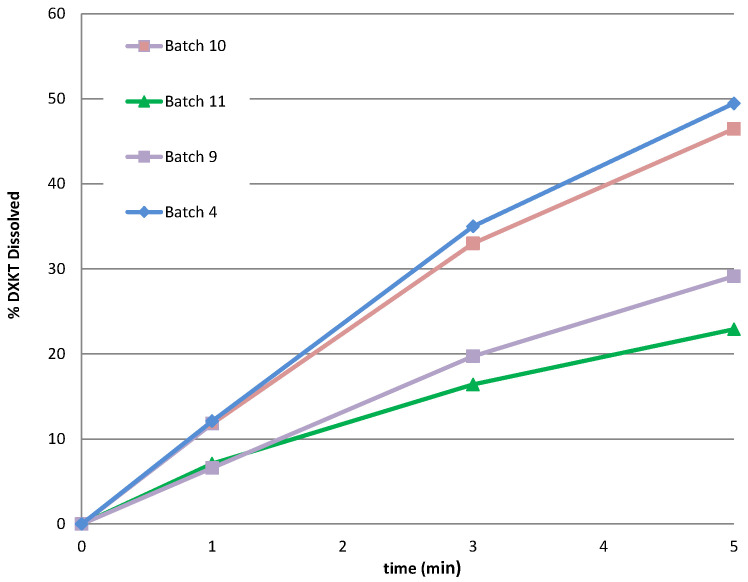
The dissolution rate of DXKT in simulated saliva from coated granules produced starting from fractions M and B with a fixed impeller speed (300 rpm) and 20% or 25% of GDS.

**Figure 11 pharmaceutics-16-00165-f011:**
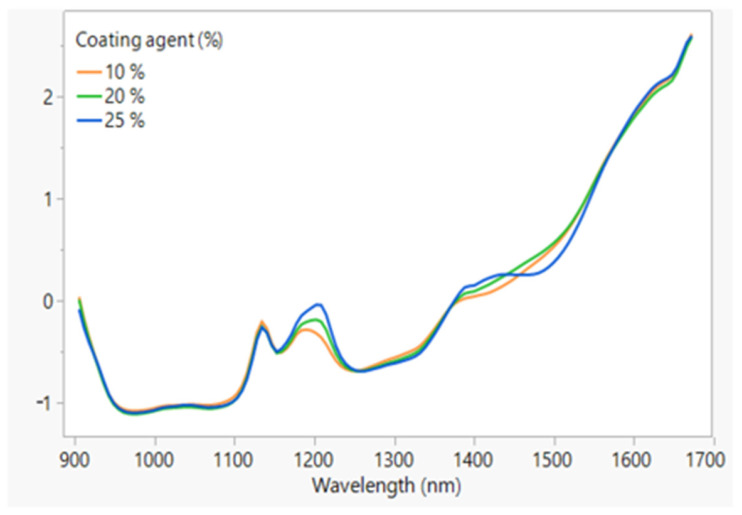
The NIR spectra of the coated granules obtained from the M fraction of the uncoated granules at a 300 rpm impeller speed and different % of GDS.

**Figure 12 pharmaceutics-16-00165-f012:**
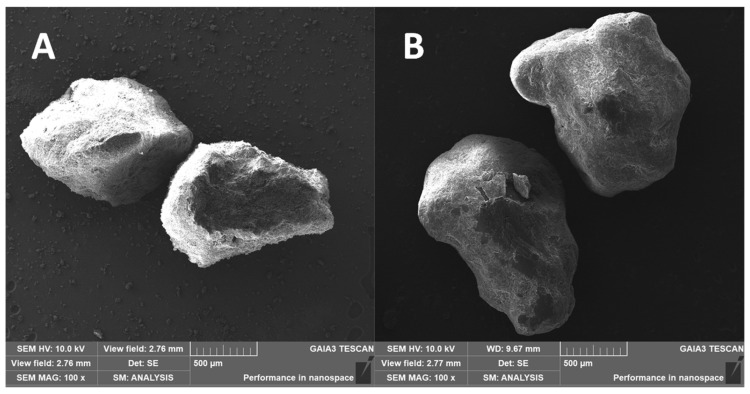
SEM micrographs at 100× magnification of uncoated granules (**A**) and of coated granules (**B**) of the Big granulometric fraction.

**Figure 13 pharmaceutics-16-00165-f013:**
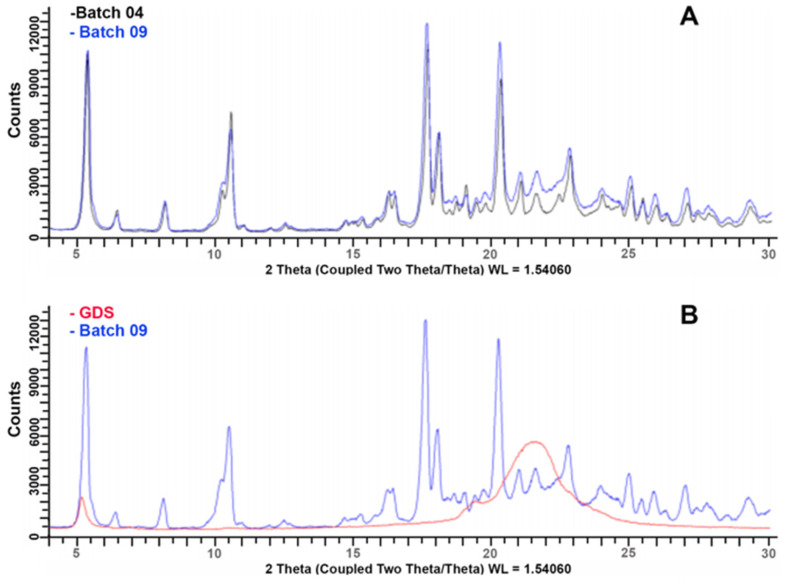
XRPD spectra of DXKT-coated granules obtained with an impeller speed of 300 rpm and 20% GDS starting from the fractions M (batch 04) and B (batch 09) (**A**) and of GDS (**B**).

**Figure 14 pharmaceutics-16-00165-f014:**
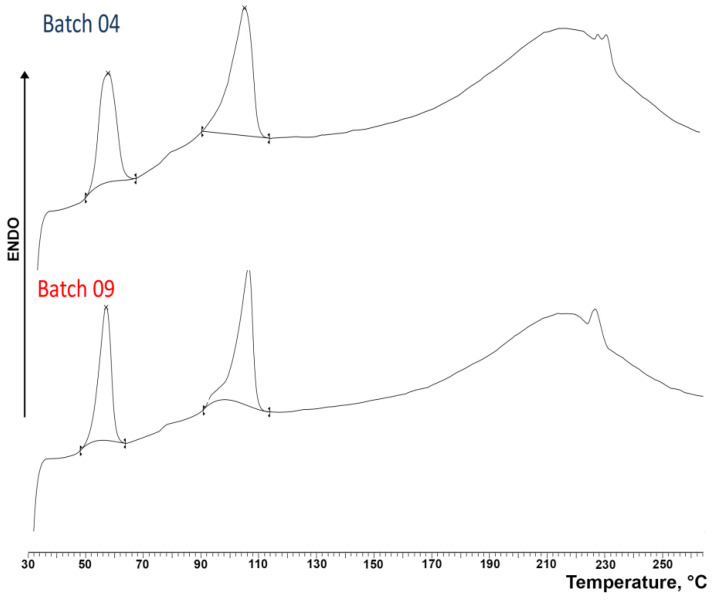
DSC curves of DXKT-coated granules obtained with an impeller speed of 300 rpm and 20% GDS starting from the granulometric fractions M (batch 04) and B (batch 09).

**Figure 15 pharmaceutics-16-00165-f015:**
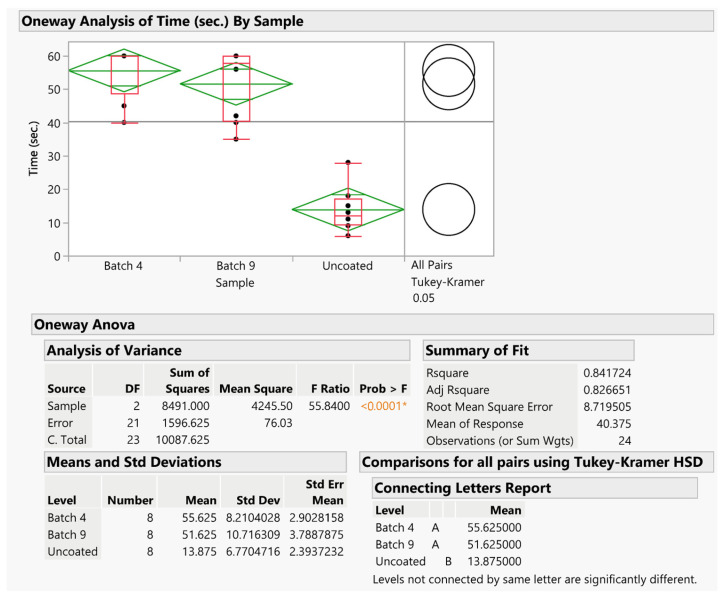
Statistical analysis with ANOVA and the Tukey–Kramer test of the data obtained from the human panel test. The asterisk indicates the statistical significance of the model; the orange color is used for *p* values lower than 0.1.

**Table 1 pharmaceutics-16-00165-t001:** Experimental plan provided by the DoE (5 experiments, including 1 center point at intermediate levels of the variables) and the obtained responses (time to reach the GDS softening temperature of 42 °C and % of DXKT dissolved in simulated saliva at 1 min).

Batch	Impeller Speed (rpm)	GDS %	Time to Reach 42 °C (min)	% Dissolved at 1 min
01	300	10	43.8	41.5
02	450	15	20.6	35.6
03	600	20	9.4	44.1
04	300	20	45.4	12.1
05	600	10	11.1	60.9

**Table 2 pharmaceutics-16-00165-t002:** Time to reach the GDS softening temperature of 42 °C and % if DXKT dissolved in simulated saliva at 1 min from coated granules obtained from fraction M at a fixed impeller speed (450 rpm) and various GDS %.

Batch	Impeller Speed (rpm)	GDS %	Time to Reach 42 °C (min)	% Dissolved at 1 min
02	450	15	20.6	35.6
06	450	20	16.5	35.3
07	450	25	15.2	34.6

**Table 3 pharmaceutics-16-00165-t003:** Time to reach the GDS softening temperature of 42 °C and % of DXKT dissolved in simulated saliva at 1 min from the coated granules obtained from the three granulometric fractions S, M, and B under the optimal experimental conditions.

Batch	Fraction	Impeller Speed (rpm)	GDS %	Time to Reach 42 °C (min)	% Dissolved at 1 min
08	S	300	20	101.3	87.9
04	M	300	20	45.4	12.1
09	B	300	20	34.6	7.1

**Table 4 pharmaceutics-16-00165-t004:** Time to reach the GDS softening temperature of 42 °C and % of DXKT dissolved in simulated saliva at 1 min from the coated granules produced with the M and B fractions of the granules and 25% of GDS (impeller speed 300 rpm).

Batch	Fraction	GDS %	Time to Reach 42 °C (min)	% Dissolved at 1 min
04	M	20	45.4	12.1
09	B	20	34.6	7.1
10	M	25	40.8	11.8
11	B	25	32.0	6.6

**Table 5 pharmaceutics-16-00165-t005:** Values of the specific surface area (SSA) obtained by BET analysis of the three uncoated granulometric fractions (S, M, and B) and their corresponding coated granules.

Uncoated Granules		Coated Granules	
Fraction	(m^2^/g)	Batch	(m^2^/g)
S	0.8	08	0.7
M	0.6	04	0.6
B	0.5	09	0.5

**Table 6 pharmaceutics-16-00165-t006:** The results of the human panel test are expressed as the mean residence time in the oral cavity before the onset of the burning sensation due to DKXT of the tested samples of the coated and uncoated granules.

Batch	Mean Residence Time in Oral Cavity before Burning Sensation (s)	Standard Error of the Mean
04	51.6	3.8
09	55.6	2.9
Uncoated granules	13.9	2.4

## Data Availability

Data are contained within the article.

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
