# Peer review of "Taste Masking of Dexketoprofen Trometamol Orally Disintegrating Granules by High-Shear Coating with Glyceryl Distearate"

_pharmaceutics, 2024, doi:10.3390/pharmaceutics16020165_

Round 1
Reviewer 1 Report
Comments and Suggestions for Authors
Dear Authors,
thank you for this manuscript. Interesting one, however, asa reviewer I had some comments. Please check them and provide the replies.
Kind regards !
Reviewer

Reviewer 2 Report
Comments and Suggestions for Authors
1. In Figure 10, the Y axis reduce the tick marks to 60.
2. In Fig. 1 and Fig. 12 should indicate the marker (scale).
3. The authors are in the process of optimising process variables, but it is necessary to convince of the importance of the selected process variables.
4. In Table 1, specify the units of time to 42°C.
5. Indicate where the centre point is in table 1?
6. Explain the need to perform a one-factor design experiment when the GDS changes from 15 to 25% if this has already been considered in the D-optimal design (Table 1).
7. Explain the necessity of performing a one-factorial design experiment with parameters on granules of different sizes if it was possible to perform a full factorial design taking into account granule size and impeller speed (rpm) and GDS (%). The same comment refers to the experiment shown in Table 4.
8. Explain why D-optimal design is used instead of response surface methodology or Taguchi design for experimental design.
Round 2
Reviewer 1 Report
Comments and Suggestions for Authors
Dear Authors,
Thank you for considering my comments. However, to be honest, I still stand by some of my original observations. Nonetheless, the manuscript is significantly improved compared to its initial version.
Kind regards !
Reviewer